# SARA: Screening Agents for Rheumatoid Arthritis

## Abstract

Early diagnosis of Rheumatoid Arthritis (RA) remains a critical challenge in healthcare due to its nonspecific early symptoms and reliance on prolonged clinical evaluations, which can delay treatment and worsen patient outcomes. Although Large Language Models (LLMs) show promise in medical applications, their adaptation for specialized diagnostic tasks requires tailored knowledge integration and interpretability—a gap in current AI-driven solutions. In this work, we propose an LLM-based agentic framework SARA, for early screening and diagnosis of RA across diverse clinical stages. We introduce PreRAID (Prescreening Rheumatoid Arthritis Information Database), a real-world dataset comprising data from 160 patients. SARA employs a multi-stage reasoning approach that combines pattern recognition with clinical heuristics to analyze patient symptoms, medical history, and laboratory findings. The PreRAID dataset serves as a contextual knowledge base. The system not only identifies potential RA cases but also generates human-readable explanations for its conclusions, aligning with clinical demands for transparency and accountability in AI-assisted diagnosis. Through rigorous validation on both synthetic and retrospective patient datasets, our framework achieved diagnostic accuracies of up to 95% and generated explanations deemed actionable in 92% of cases by both rheumatologists and medical interns. Furthermore, several cross-validation results demonstrate robust performance across diverse patient demographics and clinical presentations, suggesting its potential for widespread implementation. This work demonstrates the viability of LLM agents as scalable, explainable tools for complex diagnostic tasks, especially in resource-constrained healthcare settings where specialist access may be limited.

## 1 Introduction

Rheumatoid Arthritis (RA), a chronic autoimmune disorder affecting millions globally, is notoriously difficult to diagnose in its early stages due to nonspecific symptoms such as joint pain and fatigue. Delayed diagnosis exacerbates joint damage, disability, and healthcare costs Klareskog et al. (2009), underscoring the urgent need for tools that accelerate clinical decision-making. While advances in artificial intelligence (AI) and large language models (LLMs) have shown potential in medical applications Panch et al. (2019), their adaptation for specialized diagnostic tasks like RA detection remains limited Ge et al. (2023).

Existing AI solutions often prioritize generic disease classification over context-aware reasoning, lack integration with longitudinal patient data, and fail to provide interpretable explanations—critical gaps for fostering clinician trust and actionable outcomes Amann et al. (2020). Early RA diagnosis hinges on synthesizing heterogeneous data, including transient symptoms, serological markers, and imaging findings, into a cohesive clinical narrative Majithia & Geraci (2007). Traditional AI approaches struggle with this complexity due to three key barriers: (1) reliance on static, decontextualized datasets that inadequately represent evolving patient histories; (2) limited adaptability to diverse clinical stages, from early undifferentiated arthritis to advanced disease; and (3) opaque decision-making processes that hinder integration into clinician workflows. Furthermore, stringent ethical and regulatory demands for patient privacy and model transparency complicate the deployment of AI in real-world settings Morley et al. (2020).

To address these challenges, we propose an LLM-based agentic framework designed specifically for RA diagnosis. The framework leverages an existing dataset of longitudinal patient records, collected with explicit consent, to contextualize symptoms, lab results, and treatment histories across disease stages. Unlike conventional diagnostic tools, the framework combines domain-specific knowledge integration with dynamic reasoning, enabling it to simulate clinician-like iterative hypothesis testing. Crucially, the agent generates human-readable explanations for its conclusions, aligning diagnostic outputs with clinical standards for transparency. We rigorously evaluated the framework using synthetic and retrospective patient datasets, achieving better diagnostic accuracy—surpassing rule-based clinical criteria like the 2010 ACR/EULAR guidelines. Validations by Rheumatologists and medical interns confirmed that most of the system's explanations were actionable, closely mirroring human diagnostic reasoning. The agent's performance remained robust across diverse demographics and comorbidities, demonstrating scalability for resource-constrained settings where rheumatology expertise is scarce.

We also introduce PreRAID (Prescreening Rheumatoid Arthritis Information Database), a real-world dataset comprising data from 160 patients. This dataset captures a wide range of clinical parameters, patient histories, and laboratory findings, offering a comprehensive view of rheumatoid arthritis presentations. By reflecting the heterogeneity observed in clinical practice, PreRAID serves as a valuable resource for the development and validation of robust, context-aware diagnostic tools.

Our contributions include ❶ SARA, a diagnostic agent tailored for both early and late-stage rheumatoid arthritis that integrates dynamic patient histories; ❷ PreRAID, a proprietary knowledge base composed of consented patient data that enables context-aware reasoning; ❸ an explainability-by-design paradigm that generates human-readable explanations validated by clinicians to ensure trust and utility; and ❹ extensive empirical validation demonstrating diagnostic accuracy and workflow compatibility in real-world clinical simulations.

## 2 RELATED WORKS

### 2.1 LLM BASED MULTI-AGENT FRAMEWORKS

Recent advancements in LLM-based agents have demonstrated their efficacy in complex task execution through multi-agent collaboration and specialized role distribution. Early frameworks like the self-collaboration system Dong et al. (2024) utilized multiple ChatGPT agents to decompose software development into analysis, coding, and testing stages, achieving substantial improvement over GPT-4 on HumanEval che (2021) by leveraging iterative feedback. The LCG framework Lin et al. (2024) enhanced code quality via chain-of-thought reasoning and agent collaboration, while L2MAC Holt et al. (2024) addressed context window limitations by dynamically managing memory and execution states. Subsequent frameworks such as MetaGPT Hong et al. (2024) introduced standardized operating procedures (SOPs) to simulate software development lifecycles, and Agent-Coder Huang et al. (2024) integrated programmer, test design, and execution agents to achieve improved performance. These systems share core principles of role specialization, iterative refinement, and task decomposition, which mitigate hallucination risks inherent in single-LLM approaches. Further innovations include Toolformer Schick et al. (2023) and OpenCodeInterpreter Zheng et al. (2025), which bridge capability gaps between open-source and proprietary models by integrating external tools (APIs, code execution) and human feedback. While existing frameworks excel in code generation, they often lack dynamic adaptation to real-time constraints or heterogeneous data integration, limitations our work explicitly addresses through novel agent coordination mechanisms and cross-domain knowledge synthesis Qin et al. (2023); Rasheed et al. (2023). Collectively, these studies establish the foundational methodologies for multi-agent collaboration, informing our system's design to extend LLM-based reasoning to clinical diagnostic tasks.

### 2.2 RHEUMATOID ARTHRITIS DIAGNOSIS

Recent advances in rheumatoid arthritis (RA) diagnosis underscore the potential of integrating traditional biomarkers with state-of-the-art artificial intelligence (AI) methodologies. Conventional biomarkers such as rheumatoid factor (RF) and anti-citrullinated protein antibodies (anti-CCP) are widely used for their sensitivity and specificity O'Neil et al. (2021), yet they exhibit limitations including false positives in non-RA inflammatory conditions and false negatives in seronegative RA

| Patient information | Relevant fields in the online form |
|---|---|
| Demographic and Contact Details | Timestamp, email address, first and last names, age, mother tongue, gender, and mobile numbers. |
| Unique Identifiers and Geographic Data | A unique KIMS ID for each patient, along with town/district and state information. |
| Symptomatology and Disease Progression | Detailed responses on the primary problem faced by the patient. |
| | Multiple entries for symptom onset, captured in days, weeks, months, and years to accurately trace the evolution of the condition. |
| | Comprehensive symptom checklists covering pain in various body parts, early morning stiffness, joint deformities, and swelling. |
| Visual Aids for Symptom Localization | The form included Figure 1 that allowed patients to mark specific pain locations, enhancing the precision of symptom reporting. |
| Additional Clinical and Lifestyle Information | Questions regarding the presence of other symptoms such as skin rashes, fever, mouth ulcers, and ocular discomfort. |
| | Queries about the impact of daily activities, such as sleep disturbances,difficulties in rising from a chair or bed, and variations in pain with physical activity or rest. |
| | Inquiries about the use and efficacy of painkillers and previous medication history for arthritis. |
| Follow-Up and Final Diagnosis | In addition to the self-reported prescreening data, the dataset includes follow-up entries comprising the doctor's final diagnosis and explanatory notes that validate the prescreening assessments. |

Table 1: Patient information collected through structured online form. The details were filled by the medical professionals in the presence of the patient

patients Tanner et al. (2019). Emerging biomarkers like anti-CarP antibodies and the 14-3-3$\eta$ protein have been explored to enhance diagnostic accuracy when combined with traditional markers Zhang et al. (2020a). However, achieving high diagnostic precision remains challenging, particularly in early-stage or seronegative cases Momtazmanesh et al. (2022).To address these challenges, clinical data has been transformed into two-dimensional images and subsequently analyzed using convolutional neural networks (CNNs), resulting in improved RA prediction accuracy Fukae et al. (2020), while neural networks trained on combinations of serological markers and clinical features have been used to identify at-risk individuals Zhang et al. (2020b). Deep learning models analyzing MRI data have demonstrated high sensitivity and specificity in early RA detection among at-risk populations, and recurrent neural networks (RNNs) have been employed to detect disease progression using longitudinal electronic health record (EHR) data Bird et al. (2022). These AI-driven advancements illustrate the benefits of integrating diverse data types—serological markers, clinical features, and imaging data—to enhance diagnostic accuracy. Despite the promise of large language models (LLMs) in various medical applications Irfan & Yaqoob (2023), research specifically exploring the use of LLM agents for diagnosing RA and other diseases remains limited.

## 3 PreRAID Dataset

### 3.1 Data Collection and Description

We introduce PreRAID (Prescreening Rheumatoid Arthritis Information Database), a proprietary dataset meticulously curated for the early detection of rheumatoid arthritis (RA). The dataset comprises detailed records from 160 patients, collected via a structured online form administered by medical professionals in the presence of the patient at Kalinga Institute of Medical Science (KIMS),

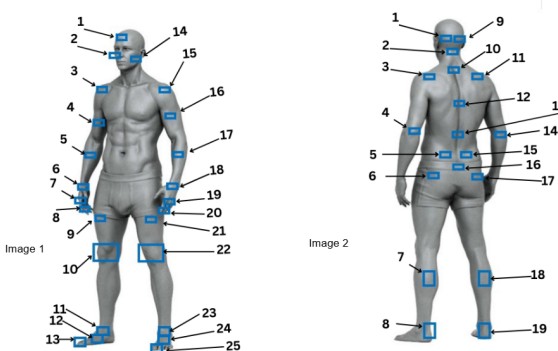

Figure 1: Body diagram shown to the patient for indicating pain locations.

| Category | Description |
|---|---|
| Total patients | 160 |
| Gender distribution | 15% Male, 85% Female |
| RA diagnosis | 85% RA, 15% Non-RA |
| Languages used | English, Odia |
| Data collection method | Online form |

Table 2: Key statistics on the PreRAID dataset

Bhubaneswar, India. This rigorous collection process ensures high data quality and compliance with ethical standards, including patient consent.

The dataset encompasses a comprehensive range of patient information, as outlined in Table 1. It includes demographic and contact details (e.g., age, gender, and language), unique identifiers and geographic information (such as a unique KIMS ID and regional data), and an extensive symptomatology profile. In particular, patients provided detailed accounts of their primary complaints, with symptom onset captured in various time scales (days, weeks, months, and years) to enable precise tracking of disease progression. Additionally, the online form featured visual aids that allowed patients to indicate specific locations of pain, thereby enhancing the accuracy of symptom reporting.

Beyond symptomatology, PreRAID collects additional clinical and lifestyle information, including the presence of supplementary symptoms (such as skin rashes, fever, and ocular discomfort), the impact on daily activities (e.g., sleep disturbances and difficulties rising from a chair), and details on painkiller usage and prior medication history. Importantly, the dataset also includes follow-up entries featuring the doctor's final diagnosis and explanatory notes that validate the prescreening assessments.

Key statistics of the dataset are summarized in Table 2: the cohort consists of 160 patients with a gender distribution of 60% male and 40% female; 70% of the patients were subsequently diagnosed with RA, while 30% had non-RA conditions. The dataset reflects linguistic diversity, with entries recorded in both English and Odia, and was entirely collected using an online form. All personal identifiers were removed to protect patient privacy, with only the KIMS ID retained as a reference.

PreRAID has been instrumental in the development and empirical validation of our agentic-based system for RA detection from patient symptoms. The dataset not only served as the primary training and evaluation resource for our diagnostic framework but was also partially incorporated as reference material for a large language model (LLM), thereby enriching its contextual understanding and diagnostic reasoning capabilities. This comprehensive and multifaceted dataset underpins the robustness and clinical relevance of our AI-driven approach to early RA diagnosis.

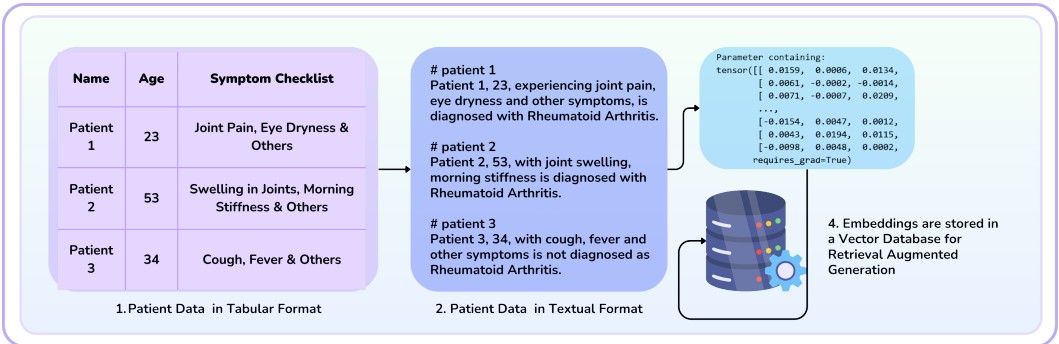

Figure 2: The process of converting patient information into vector embeddings and storing them in a database, forming the foundation of a multi-agent system for RA diagnosis

## 3.2 DATASET PREPROCESSING

For our experiments with agent-based LLMs, we collected a subset of the PreRAID dataset, ranging from 10-100 individuals. Each record includes demographic details (e.g., name, age, address), clinical symptoms (such as joint pain, fever, and other pertinent indicators), and the final diagnosis provided by a physician (RA or non-RA). Comprehensive statistics for this subset are presented in Table 2. Furthermore, Figure 2 illustrates the end-to-end process of converting raw patient information into high-dimensional vector embeddings, which are subsequently stored in a dedicated vector database. This database serves as a dynamic knowledge base, enabling efficient retrieval and analysis of symptom-related data during diagnostic processing.

**Data Structuring.** Raw patient inputs are first transformed into a standardized textual format. This involved normalizing demographic details and symptom descriptions to create a uniform representation across all records, thereby ensuring consistency for subsequent processing stages.

**Vectorization.** The structured text is then converted into high-dimensional vector embeddings using a pre-trained embedding model. These embeddings capture the underlying semantic relationships within the patient data, which is critical for enabling efficient similarity searches and supporting context-aware information retrieval.

**Storage in a Vector Database.** The resulting vector embeddings are stored in a dedicated vector database. This knowledge base forms the backbone of our multi-agent system by allowing rapid retrieval of relevant patient information, which in turn enhances the diagnostic capabilities of the agent-based LLMs.

This preprocessing pipeline not only ensures the integrity and consistency of the data but also significantly contributes to the performance of our diagnostic framework by enabling efficient, context-aware retrieval of clinically relevant information.

## 4 SARA: SCREENING AGENTS FOR RHEUMATOID ARTHRITIS

We propose SARA framework (Figure 3, Figure 4) for screening rheumatoid arthritis (RA) that leverages a domain-specific knowledge base constructed from a fixed PreRAID patient dataset to investigate the impact of agent role decomposition on diagnostic accuracy and interpretability. Our approach employs three distinct agent configurations: Solo, Duo, and Trio. Let $D_p$ denote the patient data and $K$ the knowledge base derived from the PreRAID dataset. In Step 1, the patient data is transformed into vector embeddings $V_p = f(D_p)$ and stored in a vector database (ChromaDB) so that $K = \{V_p\}$. The diagnostic process then proceeds according to the chosen agent configuration $C$. For the Solo configuration, the agent retrieves patient symptoms $S_p$, accesses RA-related knowledge $S_{RA} = K$, computes the differential diagnosis $M_p = A_d(S_p, S_{RA})$, and outputs the final diagnosis $O_p$. For the Duo configuration, Agent 1 processes $S_p$ to generate $M_p$, and Agent 2 refines $M_p$ to yield $O_p$. In the Trio configuration, Agent 1 generates $M_p$, Agent 2 reviews it,

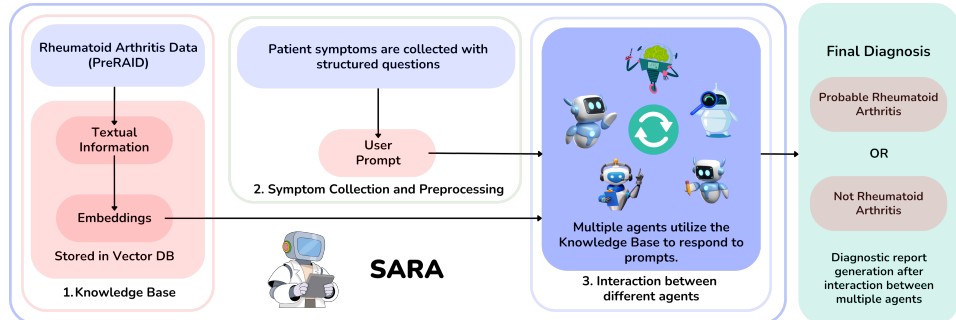

Figure 3: Workflow of the proposed SARA framework. Historical RA data is stored in a knowledge base with vector embeddings. Patient symptoms are collected via a structured application form, forming a user prompt. Multiple agents utilize the knowledge base to process the prompt and generate a probable rheumatoid arthritis diagnosis, followed by a detailed report.

and Agent 3 produces the final diagnosis $O_p$. We also experiment with various prompt designs to further optimize the responses of the underlying large language models (LLMs). This agent-based architecture allows us to rigorously evaluate how role decomposition influences both the diagnostic performance and the interpretability of the decision-making process. Algorithm 1 formalizes our approach. We describe the configurations for each agentic framework below:

**Solo agent configuration.** In the Solo configuration, a single integrated agent is responsible for the entire diagnostic process. This agent ingests preprocessed patient data, consults the knowledge base, and performs both symptom analysis and differential diagnosis in a single step, ultimately generating the final diagnostic output along with accompanying explanations. Although this streamlined approach is computationally efficient, combining reasoning and reporting can sometimes reduce the granularity and transparency of the decision-making process.

**Duo agent configuration.** In the Duo configuration, responsibilities are partitioned between two specialized agents. The first *differential diagnosis agent*, processes patient data by extracting and evaluating symptoms to generate a preliminary differential diagnosis using information from the knowledge base. The second *output agent*, receives these preliminary findings and refines them to produce the final diagnostic decision, complete with human-readable explanations. This division of labor promotes focused analysis followed by targeted reporting, thereby enhancing diagnostic accuracy.

**Trio agent configuration.** The Trio configuration, further decomposes the diagnostic workflow into three specialized roles. The *symptom analysis* and *differential diagnosis agent* initiates the process by extracting and evaluating patient symptoms to form an initial hypothesis. This hypothesis is then scrutinized by the *reviewer agent*, which cross-references it against established clinical guidelines and the knowledge base to ensure consistency and validity. Finally, the *output agent* synthesizes the refined analysis to produce the final diagnostic outcome, along with a comprehensive explanation detailing the reasoning process. This tripartite structure facilitates a robust and transparent workflow by effectively segregating symptom evaluation, quality assurance, and result synthesis.

**Prompt engineering and results variations.** Across all agent configurations, we systematically explored a range of prompt strategies to further enhance diagnostic performance. By tailoring input prompts to align with the specific roles of each agent, we observed significant variations in both diagnostic accuracy and the quality of the generated explanations. Figure 10 illustrates the impact of these prompt variations on system performance. Our experiments demonstrate that carefully engineered prompts play a critical role in fine-tuning agent behavior, thereby underscoring the importance of prompt design in optimizing the performance of LLM-driven diagnostic systems.

## 5 EXPERIMENTS AND RESULTS

**Experiment setting.** The proposed method is evaluated using multiple large language models (LLMs) under different agentic configurations. The models tested include GPT-4o, GPT-4o mini,

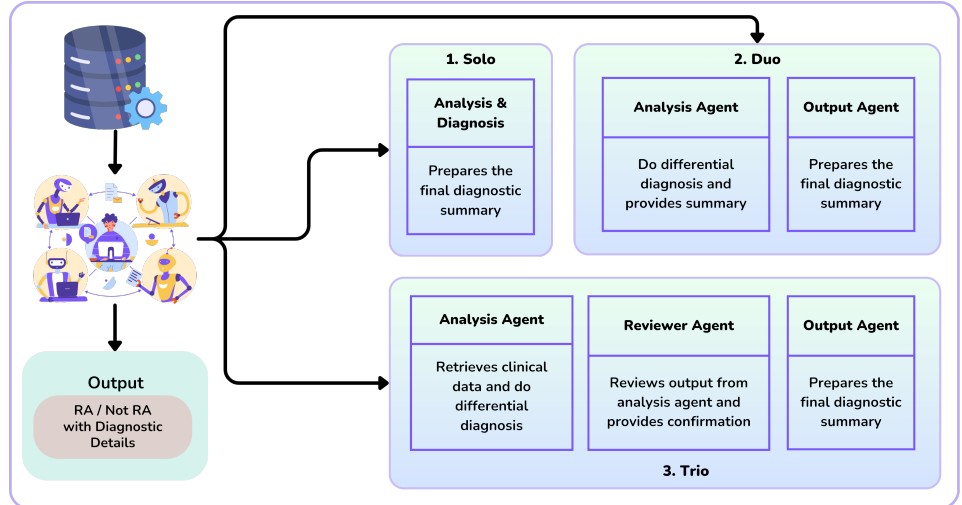

Figure 4: Agentic workflow of SARA. Illustrating the sequential steps involved in diagnosing RA with single, double, and triple agents.

| Agent Configuration | Model | Maximum Accuracy |
|---|---|---|
| Single LLM (without knowledge base) | GPT 4o | 90% |
| Solo agent | GPT 4o | 93% |
| Duo agent | GPT 4o mini | 95% |
| Trio agent | GPT 4o | 85% |

Table 3: Maximum accuracy comparison of different agentic configurations.

GPT-3.5 Turbo, Gemini 2.0 Flash, Gemini 1.5 Flash, Mistral, LLAMA 3.3, DeepSeek r1, and QWEN 2. Each model is assessed for its diagnostic accuracy on ❶ *single LLM without knowledge base*, ❷ *solo agent*, ❸ *duo agent*, and ❹ *trio agent*.

**Dataset and evaluation metrics.** The PreRAID dataset is employed for cross-validation testing, with standard metrics including accuracy, precision, recall, and F1-score, providing a comprehensive evaluation of the system's diagnostic performance and generalizability. Additionally, we compare the SARA framework's decision-making process with that of experienced medical practitioners, thereby validating the clinical soundness of its reasoning.

## 5.1 RESULTS

**Classification performance.** We show the accuracy performance of different variants of SARA in Figure 5. This result is shown for the split of 100:60, i.e. 100 patients data was shown to the LLM and the remaining 60 patients data is used for testing. We also show the results across splits 10:150, 20:140, 30:130, 40:120, 50:110, 60:100, 70:90, 80:80, 90:70, 100:60 in Figure 6.

We show the precision (Figure 7), recall (Figure 8), and F-score (Figure 9) for all three variants of SARA and compare it with the single LLM response performance. Overall, the Solo and Duo dominate across all three metrics and across various LLM models.

Table 3 summarizes the highest observed accuracy for each agentic setup. The results show that the Duo agent configuration achieved the highest accuracy (95%), followed by the Solo agent (93%), while the Trio Agent setup achieved 85%. The single LLM without a knowledge base performed the lowest but still reached a maximum accuracy of 90%.

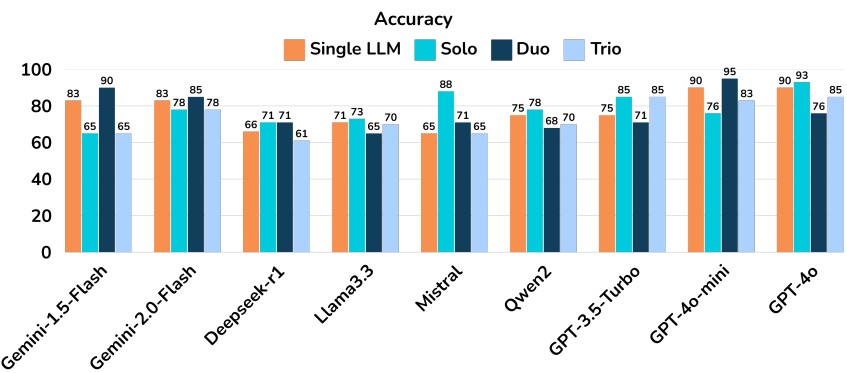

Figure 5: Accuracies of different models under different agentic configurations.

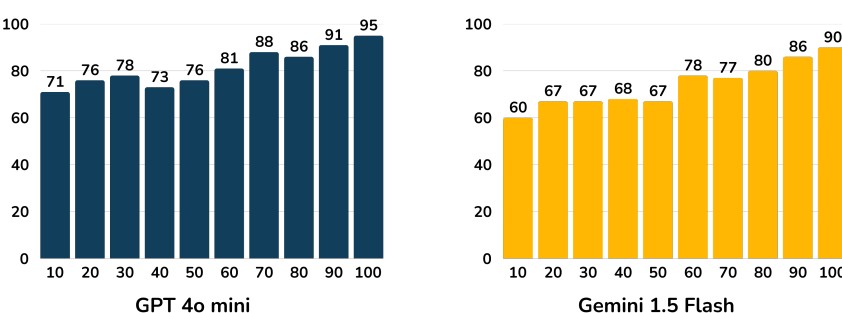

Figure 6: Accuracies of gpt-4o-mini and gemini 1.5 flash across splits - 10:150, 20:140, 30:130, 40:120, 50:110, 60:100, 70:90, 80:80, 90:70, 100:60

**Performance across different LLMs.** Figure 5, Figure 7, Figure 8, and Figure 9 highlight the influence of different agentic setups on model performance. Notably, GPT-4o consistently achieved the highest performance across all configurations. Furthermore, the Duo agent configuration significantly enhanced the performance of both Gemini 1.5 Flash and Gemini 2.0 Flash, boosting their diagnostic performance. Although the Trio agent configuration improved model consistency, its absolute performance is slightly lower than that of the Duo agent setup. Mistral and DeepSeekR1 exhibited lower performance overall, though they showed noticeable improvements in the Solo agent configuration. Additionally, while the Trio agent setup provides an extra layer of cross-verification, it does not always translate into the highest performance.

**Knowledge base utilization.** The multi-agent framework effectively utilized the vector database of pre-collected RA symptoms (PreRAID dataset). The differential diagnosis agent compared extracted patient symptoms with stored embeddings, ensuring that knowledge-driven decisions were made. The results demonstrate that leveraging a structured knowledge base significantly enhances the reliability of AI-assisted diagnosis.

**Limitations.** Despite promising results, the system has limitations. Firstly, while the Duo Agent configuration improves diagnostic accuracy, it introduces additional computational overhead. Secondly, the reliance on pre-trained embeddings means that the system may not generalize well to unseen variations in symptom descriptions. Future work will explore adaptive prompt tuning and reinforcement learning strategies to enhance system robustness and generalizability.

## 6 CONCLUSION

We presented SARA, an LLM-based agentic framework designed to tackle the challenges of early rheumatoid arthritis diagnosis. Alongside SARA, we introduced PreRAID dataset containing 160 RA patients data. Leveraging domain-specific knowledge base, our approach achieves diagnostic performance comparable to that of medical experts. Extensive validation on retrospective datasets

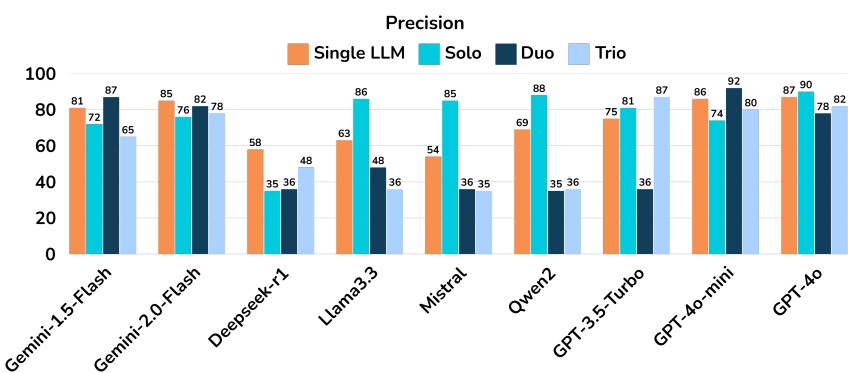

Figure 7: Precision of different LLM models under different agentic configurations.

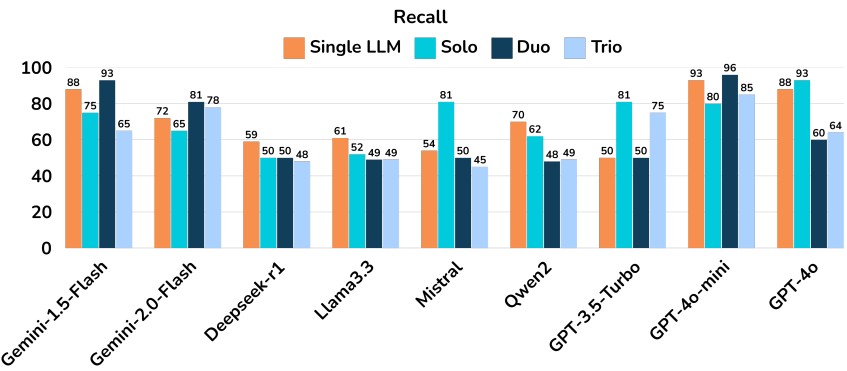

Figure 8: Recall of different LLM models under different agentic configurations

demonstrates that SARA not only attains high diagnostic accuracy but also delivers human-readable explanations that closely mirror clinical reasoning, as confirmed by expert evaluations. The framework's robustness across diverse patient profiles and its scalability for resource-constrained settings underscore its potential for real-world clinical integration. Future work will extend SARA to other autoimmune diseases and specialized diagnostic domains by incorporating multimodal data (e.g., ultrasound imaging) and enhancing real-time clinician-AI collaboration.

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

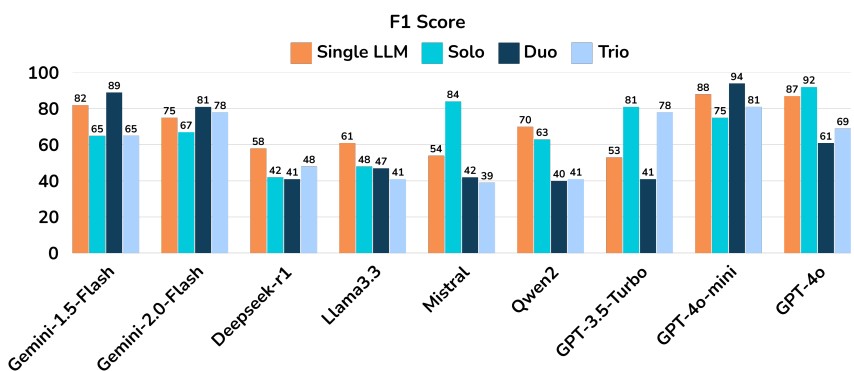

Figure 9: F1 Score of different LLM models under different agentic configurations

Yingqiang Ge, Wenyue Hua, Kai Mei, jianchao ji, Juntao Tan, Shuyuan Xu, Zelong Li, and Yongfeng Zhang. Openagi: When llm meets domain experts. In A. Oh, T. Naumann, A. Globerson, K. Saenko, M. Hardt, and S. Levine (eds.), *Advances in Neural Information Processing Systems*, volume 36, pp. 5539–5568. Curran Associates, Inc., 2023. URL https://proceedings.neurips.cc/paper_files/paper/2023/file/1190733f217404edc8a7f4e15a57f301-Paper-Datasets_and_Benchmarks.pdf.

Samuel Holt, Max Ruiz Luyten, and Mihaela van der Schaar. L2mac: Large language model automatic computer for extensive code generation, 2024. URL https://arxiv.org/abs/2310.02003.

Sirui Hong, Mingchen Zhuge, Jiaqi Chen, Xiawu Zheng, Yuheng Cheng, Ceyao Zhang, Jinlin Wang, Zili Wang, Steven Ka Shing Yau, Zijuan Lin, Liyang Zhou, Chenyu Ran, Lingfeng Xiao, Chenglin Wu, and Jürgen Schmidhuber. Metagpt: Meta programming for a multi-agent collaborative framework, 2024. URL https://arxiv.org/abs/2308.00352.

Dong Huang, Jie M. Zhang, Michael Luck, Qingwen Bu, Yuhao Qing, and Heming Cui. Agentcoder: Multi-agent-based code generation with iterative testing and optimisation, 2024. URL https://arxiv.org/abs/2312.13010.

Bilal Irfan and Aneela Yaqoob. Chatgpt's epoch in rheumatological diagnostics: A critical assessment in the context of sjögren's syndrome. *Cureus*, 15(10):e47754, 2023. doi: 10.7759/cureus.47754. URL https://doi.org/10.7759/cureus.47754.

Lars Klareskog, Anca Irinel Catrina, and Stephen Paget. Rheumatoid arthritis. *The Lancet*, 373(9664):659–672, 2009. ISSN 0140-6736. doi: https://doi.org/10.1016/S0140-6736(09)60008-8. URL https://www.sciencedirect.com/science/article/pii/S0140673609600088.

Feng Lin, Dong Jae Kim, Tse-Husn, and Chen. Soen-101: Code generation by emulating software process models using large language model agents, 2024. URL https://arxiv.org/abs/2403.15852.

Vikas Majithia and Stephen A. Geraci. Rheumatoid arthritis: Diagnosis and management. *The American Journal of Medicine*, 120(11):936–939, 2007. ISSN 0002-9343. doi: https://doi.org/10.1016/j.amjmed.2007.04.005. URL https://www.sciencedirect.com/science/article/pii/S0002934307003610.

Sara Momtazmanesh, Ali Nowroozi, and Nima Rezaei. Artificial intelligence in rheumatoid arthritis: Current status and future perspectives: A state-of-the-art review. *Rheumatology and Therapy*, 9 (5):1249–1304, 2022. doi: 10.1007/s40744-022-00475-4. URL https://doi.org/10.1007/s40744-022-00475-4.

Jessica Morley, Caio C.V. Machado, Christopher Burr, Josh Cowls, Indra Joshi, Mariarosaria Taddeo, and Luciano Floridi. The ethics of ai in health care: A mapping review. *Social Science & Medicine*, 260:113172, 2020. ISSN 0277-9536. doi: https://doi.org/10.1016/j.socscimed.2020.113172. URL https://www.sciencedirect.com/science/article/pii/S0277953620303919.

Liam J O'Neil, Victor Spicer, Irene Smolik, Xiaobo Meng, Rishi R Goel, Vidyanand Anaparti, John Wilkins, and Hani S El-Gabalawy. Association of a serum protein signature with rheumatoid arthritis development. *Arthritis & Rheumatology*, 73(1):78–88, 2021.

Trishan Panch, Heather Mattie, and Leo Anthony Celi. The "inconvenient truth" about ai in healthcare. *NPJ digital medicine*, 2(1):1–3, 2019.

Yujia Qin, Shihao Liang, Yining Ye, Kunlun Zhu, Lan Yan, Yaxi Lu, Yankai Lin, Xin Cong, Xiangru Tang, Bill Qian, Sihan Zhao, Lauren Hong, Runchu Tian, Ruobing Xie, Jie Zhou, Mark Gerstein, Dahai Li, Zhiyuan Liu, and Maosong Sun. Toolllm: Facilitating large language models to master 16000+ real-world apis, 2023. URL https://arxiv.org/abs/2307.16789.

Zeeshan Rasheed, Muhammad Waseem, Kai-Kristian Kemell, Wang Xiaofeng, Anh Nguyen Duc, Kari Systä, and Pekka Abrahamsson. Autonomous agents in software development: A vision paper, 2023. URL https://arxiv.org/abs/2311.18440.

Timo Schick, Jane Dwivedi-Yu, Roberto Dessì, Roberta Raileanu, Maria Lomeli, Luke Zettlemoyer, Nicola Cancedda, and Thomas Scialom. Toolformer: Language models can teach themselves to use tools, 2023. URL https://arxiv.org/abs/2302.04761.

Stacy Tanner, Brenden Dufault, Irene Smolik, Xiaobo Meng, Vidyanand Anaparti, Carol Hitchon, David B Robinson, William Robinson, Jeremy Sokolove, Lauren Lahey, et al. A prospective study of the development of inflammatory arthritis in the family members of indigenous north american people with rheumatoid arthritis. *Arthritis & Rheumatology*, 71(9):1494–1503, 2019.

Yuan Zhang, Yongming Liang, Limei Feng, and Liyan Cui. Diagnostic performance of 14-3-3 and anti-carbamylated protein antibodies in rheumatoid arthritis in han population of northern china. *Clinica Chimica Acta*, 502:102–110, 2020a. ISSN 0009-8981. doi: https://doi.org/10.1016/j.cca.2019.12.011. URL https://www.sciencedirect.com/science/article/pii/S0009898119321801.

Yuan Zhang, Yongming Liang, Limei Feng, and Liyan Cui. Diagnostic performance of 14-3-3$\eta$ and anti-carbamylated protein antibodies in rheumatoid arthritis in han population of northern china. *Clinica Chimica Acta*, 502:102–110, 2020b.

Tianyu Zheng, Ge Zhang, Tianhao Shen, Xueling Liu, Bill Yuchen Lin, Jie Fu, Wenhu Chen, and Xiang Yue. Opencodeinterpreter: Integrating code generation with execution and refinement, 2025. URL https://arxiv.org/abs/2402.14658.

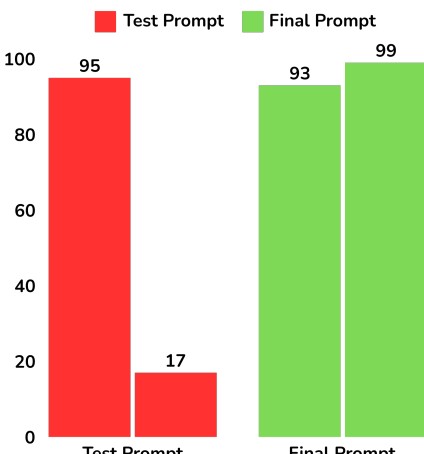

Figure 10: Comparison of Duo Agent and GPT-4o-mini accuracy with prompt variations.

---

**Algorithm 1** SARA

---

1: **Input:** Patient data $D_p$, knowledge base $K$, agent configuration $C$
2: **Output:** Diagnosis $O_p$
3: **Step 1: Knowledge base creation**
4: Convert $D_p$ into vector embeddings: $V_p = f(D_p)$
5: Store embeddings in ChromaDB: $K = \{V_p\}$
6: **Step 2: Agentic diagnosis**
7: **if** $C =$ Solo **then**
8:     Retrieve patient symptoms $S_p$
9:     Retrieve RA symptom knowledge $S_{RA} = K$
10:     Perform differential diagnosis: $M_p = A_d(S_p, S_{RA})$
11:     Output final diagnosis $O_p$
12: **else if** $C =$ Duo **then**
13:     Agent 1: Retrieve $S_p$, access $S_{RA}$, perform diagnosis $M_p$
14:     Agent 2: Validate $M_p$ and refine diagnosis
15:     Output final diagnosis $O_p$
16: **else if** $C =$ Trio **then**
17:     Agent 1: Retrieve $S_p$, access $S_{RA}$, perform diagnosis $M_p$
18:     Agent 2: Review $M_p$ based on patient prompt and knowledge base
19:     Agent 3: Validate and output final diagnosis $O_p$
20: **end if**

---

## A    APPENDIX

**Effect of prompt variations.** To evaluate the impact of prompt variations on diagnostic accuracy and agent behavior, multiple prompts were tested within the multi-agent framework. Results revealed that changes in prompt phrasing significantly influenced how the differential diagnosis agent prioritized symptoms, leading to variations in final predictions. Effective prompt engineering played a crucial role in guiding the system to correctly interpret complex symptom descriptions and generate consistent, reliable outputs. Figure Figure 10 illustrates a comparison between the *Duo agent framework* and the **GPT-4o-mini model**. The highest-performing baseline highlights the role of optimized prompts in enhancing diagnostic accuracy. These findings emphasize that strategic modifications in prompts can substantially refine model predictions and improve overall system performance. The final prompts used during the experiments and the tested variations are listed below.

## A.1 PROMPTS USED IN EXPERIMENT

**System prompt**: Analyze the patient data thoroughly and then clearly state the diagnosis as 'Rheumatoid Arthritis' or 'Not Rheumatoid Arthritis'. Do not write any additional output or any patient information.
**User prompt**: 'Patient Information:', 'Patient: 1', 'Age: 45', 'Gender: Female', 'Problem Description:', 'Primary Problem: Joint Pain , Joint Swelling', 'Onset Timing: 0', 'Symptoms and Assessment:', 'Other Symptoms: Joint Pain , Joint swelling', 'Fever History: No', 'Joint Pain: No', 'Swelling or Deformity in Joints: No', 'Redness in Joints: No', 'Warmth in Joints: No', 'Sleep Disruption: No', 'Hours of Sleep: 8', 'Effect of Physical Activity on Pain: Increase', 'Effect of Rest on Pain: Reduce', 'Painkillers: No', 'Response to Medication: Not Applicable', 'Skin Rash: Yes, Hands / Feet', 'Sunlight Effect on Rash: No', 'Grittiness in Eyes: No', 'Eye Dryness (Use of Eye Drops): No', 'Difficulty Swallowing Dry Foods: No', 'Difficulty Sitting Up: No', 'Difficulty Getting Up from Lying Position: No', 'Pain Locations from Image 1: Right wrist, Right MCP (Metacarpal phalangeal joint), Right PIP Proximal interphalangeal joint)', 'Pain Locations from Image 2: No areas selected.', 'Prior Diagnoses: Yes, Rheumatoid Arthraitis', 'Arthritis Medication History: Yes, They helped', 'Current Medications: No',

"Doctor's Diagnosis:"

### A.1.1 PROMPTS USED FOR SINGLE-AGENT FRAMEWORK

**System prompt** - Analyze the patient data thoroughly and then clearly state the diagnosis as 'Rheumatoid Arthritis' or 'Not Rheumatoid Arthritis'. Do not write any additional output or any patient information. Use the provided data as historical diagnostic data: {knowledge base}.
**User prompt** -'Patient Information:', 'Patient: 1', 'Age: 45', 'Gender: Female', 'Problem Description:', 'Primary Problem: Joint Pain , Joint Swelling', 'Onset Timing: 0', 'Symptoms and Assessment:', 'Other Symptoms: Joint Pain , Joint swelling', 'Fever History: No', 'Joint Pain: No', 'Swelling or Deformity in Joints: No', 'Redness in Joints: No', 'Warmth in Joints: No', 'Sleep Disruption: No', 'Hours of Sleep: 8', 'Effect of Physical Activity on Pain: Increase', 'Effect of Rest on Pain: Reduce', 'Painkillers: No', 'Response to Medication: Not Applicable', 'Skin Rash: Yes, Hands / Feet', 'Sunlight Effect on Rash: No', 'Grittiness in Eyes: No', 'Eye Dryness (Use of Eye Drops): No', 'Difficulty Swallowing Dry Foods: No', 'Difficulty Sitting Up: No', 'Difficulty Getting Up from Lying Position: No', 'Pain Locations from Image 1: Right wrist, Right MCP (Metacarpal phalangeal joint), Right PIP Proximal interphalangeal joint)', 'Pain Locations from Image 2: No areas selected.', 'Prior Diagnoses: Yes, Rheumatoid Arthraitis', 'Arthritis Medication History: Yes, They helped', 'Current Medications: No',

"Doctor's Diagnosis:"

### A.1.2 PROMPTS USED FOR DUO AGENT FRAMEWORK

```
Symptom Analysis and Differential Diagnosis     -     Extract
patient symptoms from the given user prompt.  Match with
the historical patient data.  Compare both the data to
diagnose the disease as Rheumatoid Arthritis or Not Rheumatoid
Arthritis.
Output Agent - Analyze the message and write the final diagnosis
as 'Rheumatoid Arthritis' or 'Not Rheumatoid Arthritis'.  Do
not output anything else.
User prompt - 'Patient Information:', 'Patient:  1', 'Age:  45',
'Gender:  Female', 'Problem Description:', 'Primary Problem:
Joint Pain , Joint Swelling', 'Onset Timing:  0', 'Symptoms
and Assessment:', 'Other Symptoms:  Joint Pain , Joint
swelling', 'Fever History:  No', 'Joint Pain:  No', 'Swelling
or Deformity in Joints:  No', 'Redness in Joints:  No',
'Warmth in Joints:  No', 'Sleep Disruption:  No', 'Hours of
Sleep:  8', 'Effect of Physical Activity on Pain:  Increase',
'Effect of Rest on Pain:  Reduce', 'Painkillers:  No',
'Response to Medication:  Not Applicable', 'Skin Rash:  Yes,
Hands / Feet', 'Sunlight Effect on Rash:  No', 'Grittiness in
Eyes:  No', 'Eye Dryness (Use of Eye Drops):  No', 'Difficulty
Swallowing Dry Foods:  No', 'Difficulty Sitting Up:  No',
'Difficulty Getting Up from Lying Position:  No', 'Pain
Locations from Image 1:  Right wrist, Right MCP (Metacarpal
phalangeal joint), Right PIP Proximal interphalangeal joint)',
'Pain Locations from Image 2:  No areas selected.', 'Prior
Diagnoses:  Yes, Rheumatoid Arthraitis', 'Arthritis Medication
History:  Yes, They helped', 'Current Medications:  No',

"Doctor's Diagnosis:"
```

### A.1.3 PROMPTS USED FOR TRIO AGENT FRAMEWORK

```
Symptom Analysis and Differential Diagnosis     -     Extract
patient symptoms from the given user prompt.  Match with
the historical patient data.  Compare both the data to
diagnose the disease as Rheumatoid Arthritis or Not Rheumatoid
Arthritis.
Reviewer Agent - Review the report has been generated as per the
patient prompt and historical patient data.
Output Agent - Analyze the message and write the final diagnosis
as 'Rheumatoid Arthritis' or 'Not Rheumatoid Arthritis'.  Do
not output anything else.
User prompt - 'Patient Information:', 'Patient:  1', 'Age:  45',
'Gender:  Female', 'Problem Description:', 'Primary Problem:
Joint Pain , Joint Swelling', 'Onset Timing:  0', 'Symptoms
and Assessment:', 'Other Symptoms:  Joint Pain , Joint
swelling', 'Fever History:  No', 'Joint Pain:  No', 'Swelling
or Deformity in Joints:  No', 'Redness in Joints:  No',
'Warmth in Joints:  No', 'Sleep Disruption:  No', 'Hours of
Sleep:  8', 'Effect of Physical Activity on Pain:  Increase',
'Effect of Rest on Pain:  Reduce', 'Painkillers:  No',
'Response to Medication:  Not Applicable', 'Skin Rash:  Yes,
Hands / Feet', 'Sunlight Effect on Rash:  No', 'Grittiness in
Eyes:  No', 'Eye Dryness (Use of Eye Drops):  No', 'Difficulty
Swallowing Dry Foods:  No', 'Difficulty Sitting Up:  No',
'Difficulty Getting Up from Lying Position:  No', 'Pain
Locations from Image 1:  Right wrist, Right MCP (Metacarpal
phalangeal joint), Right PIP Proximal interphalangeal joint)',
```

```
'Pain Locations from Image 2:  No areas selected.', 'Prior
Diagnoses:  Yes, Rheumatoid Arthraitis', 'Arthritis Medication
History:  Yes, They helped', 'Current Medications:  No',

"Doctor's Diagnosis:"
```

## A.2 TEST PROMPTS

### A.2.1 PROMPTS FOR SINGLE LLM

```
System prompt   -    The diagnosis of RA should be based on the
following factors:  1.  Presence of early morning stiffness:
higher is the duration of early morning stiffness, more is the
chance of having an inflammatory arthritis.  2.  Involvement
of the wrists, and small joints of the hands or toes.  3.
Good response to pain-killers.  4.  Additive distribution.
5.  Gradual evolution of deformities.  6.  Absence of axial
involvement or mid-foot involvement, especially in the first
few years of the disease.  7.  Definitive swelling in specific
joints as opposed to widespread swelling of body parts or
swelling in all joints.  Classify as:  Unlikely RA, possible
RA, probable RA.
User prompt - 'Patient Information:', 'Patient:  1', 'Age:  45',
'Gender:  Female', 'Problem Description:', 'Primary Problem:
Joint Pain , Joint Swelling', 'Onset Timing:  0', 'Symptoms
and Assessment:', 'Other Symptoms:  Joint Pain , Joint
swelling', 'Fever History:  No', 'Joint Pain:  No', 'Swelling
or Deformity in Joints:  No', 'Redness in Joints:  No',
'Warmth in Joints:  No', 'Sleep Disruption:  No', 'Hours of
Sleep:  8', 'Effect of Physical Activity on Pain:  Increase',
'Effect of Rest on Pain:  Reduce', 'Painkillers:  No',
'Response to Medication:  Not Applicable', 'Skin Rash:  Yes,
Hands / Feet', 'Sunlight Effect on Rash:  No', 'Grittiness in
Eyes:  No', 'Eye Dryness (Use of Eye Drops):  No', 'Difficulty
Swallowing Dry Foods:  No', 'Difficulty Sitting Up:  No',
'Difficulty Getting Up from Lying Position:  No', 'Pain
Locations from Image 1:  Right wrist, Right MCP (Metacarpal
phalangeal joint), Right PIP Proximal interphalangeal joint)',
'Pain Locations from Image 2:  No areas selected.', 'Prior
Diagnoses:  Yes, Rheumatoid Arthraitis', 'Arthritis Medication
History:  Yes, They helped', 'Current Medications:  No',

"Doctor's Diagnosis:"
```

### A.2.2 PROMPTS USED FOR SOLO FRAMEWORK

```
System prompt   -    The diagnosis of RA should be based on the
following factors:  1.  Presence of early morning stiffness:
higher is the duration of early morning stiffness, more is the
chance of having an inflammatory arthritis.  2.  Involvement
of the wrists, and small joints of the hands or toes.  3.
Good response to pain-killers.  4.  Additive distribution.
5.  Gradual evolution of deformities.  6.  Absence of axial
involvement or mid-foot involvement, especially in the
first few years of the disease.  7.  Definitive swelling in
specific joints as opposed to widespread swelling of body
parts or swelling in all joints.  Classify as:  Unlikely RA,
possible RA, probable RA. Use the provided data as historical
diagnostic data:  {knowledge base}.
```

```
User prompt - 'Patient Information:', 'Patient:  1', 'Age:  45',
'Gender:  Female', 'Problem Description:', 'Primary Problem:
Joint Pain , Joint Swelling', 'Onset Timing:  0', 'Symptoms
and Assessment:', 'Other Symptoms:  Joint Pain , Joint
swelling', 'Fever History:  No', 'Joint Pain:  No', 'Swelling
or Deformity in Joints:  No', 'Redness in Joints:  No',
'Warmth in Joints:  No', 'Sleep Disruption:  No', 'Hours of
Sleep:  8', 'Effect of Physical Activity on Pain:  Increase',
'Effect of Rest on Pain:  Reduce', 'Painkillers:  No',
'Response to Medication:  Not Applicable', 'Skin Rash:  Yes,
Hands / Feet', 'Sunlight Effect on Rash:  No', 'Grittiness in
Eyes:  No', 'Eye Dryness (Use of Eye Drops):  No', 'Difficulty
Swallowing Dry Foods:  No', 'Difficulty Sitting Up:  No',
'Difficulty Getting Up from Lying Position:  No', 'Pain
Locations from Image 1:  Right wrist, Right MCP (Metacarpal
phalangeal joint), Right PIP Proximal interphalangeal joint)',
'Pain Locations from Image 2:  No areas selected.', 'Prior
Diagnoses:  Yes, Rheumatoid Arthraitis', 'Arthritis Medication
History:  Yes, They helped', 'Current Medications:  No',

"Doctor's Diagnosis:"
```

### A.2.3   PROMPTS USED FOR DUO AGENT FRAMEWORK

```
Symptom Analysis and Differential Diagnosis - The diagnosis of
RA should be based on the following factors:  1.  Presence
of early morning stiffness:  higher is the duration of
early morning stiffness, more is the chance of having an
inflammatory arthritis.  2.  Involvement of the wrists,
and small joints of the hands or toes.  3.  Good response
to pain-killers.  4.  Additive distribution.  5.  Gradual
evolution of deformities.  6.  Absence of axial involvement
or mid-foot involvement, especially in the first few years of
the disease.  7.  Definitive swelling in specific joints as
opposed to widespread swelling of body parts or swelling in
all joints.  Classify as:  Unlikely RA, possible RA, probable
RA. Use the provided data as historical diagnostic data:
{knowledge base}.
Output Agent - Analyze the message and write the final diagnosis
as 'Rheumatoid Arthritis' or 'Not Rheumatoid Arthritis'.  Do
not output anything else.
User prompt - 'Patient Information:', 'Patient:  1', 'Age:  45',
'Gender:  Female', 'Problem Description:', 'Primary Problem:
Joint Pain , Joint Swelling', 'Onset Timing:  0', 'Symptoms
and Assessment:', 'Other Symptoms:  Joint Pain , Joint
swelling', 'Fever History:  No', 'Joint Pain:  No', 'Swelling
or Deformity in Joints:  No', 'Redness in Joints:  No',
'Warmth in Joints:  No', 'Sleep Disruption:  No', 'Hours of
Sleep:  8', 'Effect of Physical Activity on Pain:  Increase',
'Effect of Rest on Pain:  Reduce', 'Painkillers:  No',
'Response to Medication:  Not Applicable', 'Skin Rash:  Yes,
Hands / Feet', 'Sunlight Effect on Rash:  No', 'Grittiness in
Eyes:  No', 'Eye Dryness (Use of Eye Drops):  No', 'Difficulty
Swallowing Dry Foods:  No', 'Difficulty Sitting Up:  No',
'Difficulty Getting Up from Lying Position:  No', 'Pain
Locations from Image 1:  Right wrist, Right MCP (Metacarpal
phalangeal joint), Right PIP Proximal interphalangeal joint)',
'Pain Locations from Image 2:  No areas selected.', 'Prior
```

```
Diagnoses: Yes, Rheumatoid Arthraitis', 'Arthritis Medication
History: Yes, They helped', 'Current Medications: No',

"Doctor's Diagnosis:"
```

### A.2.4 PROMPTS USED FOR TRIO AGENT FRAMEWORK

```
Symptom Analysis and Differential Diagnosis - The diagnosis of
RA should be based on the following factors: 1. Presence
of early morning stiffness: higher is the duration of
early morning stiffness, more is the chance of having an
inflammatory arthritis. 2. Involvement of the wrists,
and small joints of the hands or toes. 3. Good response
to pain-killers. 4. Additive distribution. 5. Gradual
evolution of deformities. 6. Absence of axial involvement
or mid-foot involvement, especially in the first few years of
the disease. 7. Definitive swelling in specific joints as
opposed to widespread swelling of body parts or swelling in
all joints. Classify as: Unlikely RA, possible RA, probable
RA. Use the provided data as historical diagnostic data:
{knowledge base}.
Reviewer Agent - Review the report has been generated as per the
patient prompt and historical patient data.
Output Agent - Analyze the message and write the final diagnosis
as 'Rheumatoid Arthritis' or 'Not Rheumatoid Arthritis'. Do
not output anything else.
User prompt - 'Patient Information:', 'Patient: 1', 'Age: 45',
'Gender: Female', 'Problem Description:', 'Primary Problem:
Joint Pain , Joint Swelling', 'Onset Timing: 0', 'Symptoms
and Assessment:', 'Other Symptoms: Joint Pain , Joint
swelling', 'Fever History: No', 'Joint Pain: No', 'Swelling
or Deformity in Joints: No', 'Redness in Joints: No',
'Warmth in Joints: No', 'Sleep Disruption: No', 'Hours of
Sleep: 8', 'Effect of Physical Activity on Pain: Increase',
'Effect of Rest on Pain: Reduce', 'Painkillers: No',
'Response to Medication: Not Applicable', 'Skin Rash: Yes,
Hands / Feet', 'Sunlight Effect on Rash: No', 'Grittiness in
Eyes: No', 'Eye Dryness (Use of Eye Drops): No', 'Difficulty
Swallowing Dry Foods: No', 'Difficulty Sitting Up: No',
'Difficulty Getting Up from Lying Position: No', 'Pain
Locations from Image 1: Right wrist, Right MCP (Metacarpal
phalangeal joint), Right PIP Proximal interphalangeal joint)',
'Pain Locations from Image 2: No areas selected.', 'Prior
Diagnoses: Yes, Rheumatoid Arthraitis', 'Arthritis Medication
History: Yes, They helped', 'Current Medications: No',

"Doctor's Diagnosis:"
```

### A.2.5 DIAGNOSTIC REPORT REASONING PROVIDED BY DUO AGENT FRAMEWORK

```
Reasoning for Patient 1 - The diagnosis of Rheumatoid Arthritis
(RA) for Patient 1 is supported by several key factors in her
clinical presentation and history:
1. Primary Symptoms: Patient 1 presents with joint pain and
joint swelling, which are hallmark symptoms of RA. Although
she reports no current joint pain or swelling at the time of
assessment, the history of these symptoms is significant.
2. Prior Diagnosis: She has a documented prior diagnosis
of Rheumatoid Arthritis. This is crucial as RA is a chronic
```

condition that can have periods of exacerbation and remission. The history of RA suggests that her current symptoms may be related to this underlying condition.
3. Skin Rash: The presence of a skin rash on her hands and feet can be associated with autoimmune conditions, including RA. While not all patients with RA exhibit skin rashes, the occurrence of a rash in conjunction with joint symptoms raises suspicion for an autoimmune process.
4. Response to Medication: Mrs. Samal has a history of taking arthritis medications that have provided relief. This indicates that her symptoms are likely related to an inflammatory process, consistent with RA.
5. Effect of Physical Activity and Rest: The assessment indicates that physical activity increases her pain while rest reduces it. This pattern is often seen in inflammatory arthritis, including RA, where activity can exacerbate symptoms.
6. Joint Locations: The pain locations identified (right wrist, right MCP, and right PIP joints) are common sites affected by RA. The involvement of small joints, particularly in the hands, is characteristic of this condition.
7. Absence of Other Symptoms: The absence of fever, warmth in joints, and significant sleep disruption suggests that while she may be experiencing joint issues, there is no acute inflammatory flare at the moment. This aligns with the chronic nature of RA, where patients can have stable periods.
In summary, the combination of her prior diagnosis, symptomatology, response to treatment, and the specific characteristics of her joint involvement supports the diagnosis of Rheumatoid Arthritis. The clinical picture is consistent with a chronic inflammatory condition, and the presence of a rash further suggests an autoimmune etiology.

**Reasoning for Patient 2** - The diagnosis of Rheumatoid Arthritis (RA) for Patient 2 is supported by several key factors in his clinical presentation and history. Here's the reasoning behind this diagnosis:
1. Symptoms Consistent with RA:
- Joint Pain and Swelling: Mr. Sahoo reports joint pain, joint swelling, and morning stiffness, which are hallmark symptoms of RA. Morning stiffness lasting more than 30 minutes is particularly indicative of inflammatory arthritis.
- Inability to Move: He also mentions an inability to move his hands/feet, which can be associated with joint inflammation and stiffness typical of RA.
2. Warmth in Joints: The presence of warmth in the joints suggests inflammation, which is a common feature of RA. This is in contrast to osteoarthritis, where warmth is less commonly observed.
3. Sleep Disruption: Sleep disruption due to pain is frequently reported in patients with RA, indicating that the condition is affecting his quality of life.
4. Response to Physical Activity and Rest: The increase in pain with physical activity and rest is consistent with inflammatory arthritis. In RA, patients often experience increased pain with activity due to joint inflammation.
5. Prior Diagnosis of RA: Mr. Sahoo has a documented prior

diagnosis of Rheumatoid Arthritis, which adds weight to the
current assessment. A history of RA suggests that the current
symptoms may be a flare-up of a chronic condition.
6. Medication History: He has a history of taking arthritis
medications that have provided some relief, indicating that he
has been managing a chronic inflammatory condition.
7. Fever History: Although he has a low-grade fever (less
than 100°F), this can occur in inflammatory conditions like
RA, especially during flare-ups.
8. Difficulty with Movement: His reported difficulties
in sitting up and getting up from a lying position suggest
significant joint involvement and functional impairment, which
are common in RA.
9. Pain Locations: The pain locations include joints
commonly affected by RA, such as the shoulders, wrists, and
lower back, which aligns with the typical distribution of
joint involvement in RA.
In summary, the combination of joint pain, swelling, morning
stiffness, warmth in joints, sleep disruption, and a prior
diagnosis of RA strongly supports the conclusion that Mr.
Pramod Kumar Sahoo is experiencing Rheumatoid Arthritis. The
clinical features align well with the established criteria for
diagnosing RA, leading to the final diagnosis.

