# OpenReview forum: "Sara: Screening Agents for Rheumatoid Arthritis"
_ICLR.cc/2025/Workshop/AgenticAI — ICLR 2025 Workshop AgenticAI Reject_

### Official Review · Reviewer_eUKS · 2025-03-03

**Rating:** 4
**Confidence:** 4

**Review:**

The paper proposes an LLM-based agentic framework, SARA, designed for the early screening and diagnosis of Rheumatoid Arthritis (RA) across various clinical stages. Additionally, it introduces a new dataset resource, PreRAID (Prescreening Rheumatoid Arthritis Information Database), which comprises real-world data from 160 patients. The study includes extensive cross-validation experiments to evaluate the framework’s effectiveness.

Weakness:
- The gender and RA diagnosis distribution is not consistent between Table 2 and L207-208
- The paper lacks details on how patient data is transformed into vector embeddings. What pretrained model was used for encoding? What is the dimensionality of the vector embeddings? If it is one, how does the method aggregate the embeddings over all tokens?
- Table 3 is meant for comparing different agentic configurations. However, the choice of models appears inconsistent. Specifically, the Duo agent uses GPT-4o min, while other configurations use GPT-4o.
- The paper does not adequately discuss why the Trio agent underperforms compared to the Duo agent.
- As shown in Figure 5, a single LLM without a knowledge base can sometimes outperform both the Solo and Trio agents. The authors should incorporate a detailed discussion and analysis to explain these findings.
- The paper does not provide any results or analysis regarding the quality and effectiveness of the generated human-readable explanations. Since explainability is also one of the contribution of the paper, this aspect should be discussed.
- The paper should use parenthetical citations when references are not used as nouns.

---

### Official Review · Reviewer_DmNK · 2025-03-03
**Sara: Screening Agents for Rheumatoid Arthritis**

**Rating:** 8
**Confidence:** 4

**Review:**

# 1. Summary

In this paper, early Rheumatoid Arthritis (RA) diagnosis is addressed by proposing  SARA, which employs large language models (LLMs) divided into three agent roles, Solo, Duo, and Trio, to simulate the clinical reasoning process. The system diagnoses RA and provides human‐readable explanations using a new dataset, PreRAID, developed from 160 patients.

 1. **Novel Approach:**    The multi-agent design shows a significant improvement over existing attempts to deal with the clinical decision-making issues. The framework decomposes the diagnostic process into specialized roles, such that the framework acts more like a transparent, easy-to-interpret collaboration among experts than a single, opaque system.



2. **Data Contribution:**
it is well-annotated and can be used for further research.

 3. **Robust Evaluation:**
 In this paper,  experiments are performed using multiple metrics.  This robust evaluation supports the approach's validity and the importance of thoughtful, prompt engineering to boost performance.

4. **Explainability:**
 A significant advantage is the system’s ability to generate explanations that align with clinical reasoning.

# 3. Weaknesses

 1. **Limited Dataset Size**
 The study may not have fully captured the diversity of populations because of the limited 160 records in the dataset, which raise a concern about its generlization.

2. **Scalability and Overhead**
 Despite its high performance, the Duo configuration introduces additional computational complexity, which may be challenging for real-world clinical implementation.

---

### Official Review · Reviewer_Artm · 2025-03-04
**Sara: Screening Agents for Rheumatoid Arthritis**

**Rating:** 5
**Confidence:** 4

**Review:**

The paper introduces SARA, an LLM-based multi-agent framework for early Rheumatoid Arthritis (RA) diagnosis, achieving high accuracy (95%) and producing clinician-approved explanations (92%) using the PreRAID dataset (160 patients). While the paper is well-written, there are several key concerns. First, the dataset is small and geographically constrained, limiting generalizability. Second, the Duo/Trio agent configurations introduce significant computational overhead, raising concerns about scalability. Additionally, the fixed knowledge base restricts adaptability to new medical insights, and the methodology for obtaining contextual embeddings lacks clarity—specifically, which pre-trained embedding model was used (Lines 246/247)? From a framework perspective, the approach lacks novelty, as it primarily builds on existing multi-agent paradigms without introducing fundamental improvements. Additionally, in Table 3, why was the GPT-4o Mini result reported for the Duo agent configuration, whereas the GPT-4o result has been reported for the other settings? Furthermore, Section 5.1 lacks in-depth analysis, primarily summarizing results from tables and plots rather than offering insights into why specific agent configurations outperform others or how ablations impact performance. A more thorough discussion of the reasoning behind model superiority, failure cases, and comparative analysis is needed to enhance the scientific depth and clarity of the experimental results.

---

### Decision · Program_Chairs · 2025-03-05

Reject